# Clinical Significance of PD-L1 Status in Circulating Tumor Cells for Cancer Management during Immunotherapy

**DOI:** 10.3390/biomedicines11061768

**Published:** 2023-06-20

**Authors:** Areti Strati, Panagiota Economopoulou, Evi Lianidou, Amanda Psyrri

**Affiliations:** 1Analysis of Circulating Tumor Cells, Laboratory of Analytical Chemistry, Department of Chemistry, National and Kapodistrian University of Athens, 15771 Athens, Greece; lianidou@chem.uoa.gr; 2Department of Internal Medicine, Section of Medical Oncology, National and Kapodistrian University of Athens, Attikon University Hospital, 12462 Athens, Greece; panagiota_oiko@hotmail.com (P.E.); psyrri237@yahoo.com (A.P.)

**Keywords:** PD-L1, CTCs, immunotherapy, immune checkpoint inhibitors

## Abstract

The approval of monoclonal antibodies against programmed death-ligand 1 (PD-L1) and programmed cell death protein (PD1) has changed the landscape of cancer treatment. To date, many immune checkpoint inhibitors (ICIs) have been approved by the FDA for the treatment of metastatic cancer as well as locally recurrent advanced cancer. However, immune-related adverse events (irAEs) of ICIs highlight the need for biomarker analysis with strong predictive value. Liquid biopsy is an important tool for clinical oncologists to monitor cancer patients and administer or change appropriate therapy. CTCs frequently express PD-L1, and this constitutes a clinically useful and non-invasive method to assess PD-L1 status in real-time. This review summarizes all the latest findings about the clinical significance of CTC for the management of cancer patients during the administration of immunotherapy and mainly focuses on the assessment of PD-L1 expression in CTCs.

## 1. Introduction

Minimally invasive liquid biopsies allow the analysis of tumor elements, such as circulating tumor cells (CTCs) and circulating tumor DNA (ctDNA) in body fluids, mainly blood. The liquid biopsy concept launched approximately 10 years ago, has opened new horizons in cancer prevention, diagnosis, early identification of tumor recurrence, molecular characterization of tumors, monitoring of response to treatment, and detection of resistance mechanisms [1]. Clinical applications of CTCs have gained enormous attention over the past few years, despite many limitations in CTC capture by current methodologies [2]. Indeed, CTCs are being evaluated as predictive biomarkers since CTC analysis provides rapid, tumor-specific information that can be repeatedly accessible during follow-up and enables monitoring of response to treatment and early identification of resistance mechanisms.

The Food and Drugs Administration (FDA)-cleared CellSearch platform represents the gold standard for CTC detection and enumeration in the bloodstream [3]. CTC enumeration could provide the primary detection of metastatic cancer in contrast to radiological tests [4]. This creates a huge prospect, especially in the field of early cancer diagnosis. Moreover, one of the main advantages of CTC count using the CellSearch platform is that it enables the stratification of metastatic breast cancer patients (MBC) [5]. Additionally, the implementation of CellSearch allows the detection of CTC clusters that have high metastatic potential and hold great promise for metastatic breast cancer therapy [6].

In recent years, the promising results yielded from the development of CTCs in targeted therapies have paved the way for the implementation of CTCs into the domain of immunotherapy. The most important biomarker for treatment decision-making in the era of immunotherapy is programmed death-ligand 1 (*PD-L1*) expression, which is assessed by immunohistochemistry in tumor specimens [7]. However, several issues, such as intra-and inter-tumor heterogeneity and expression of *PD-L1* in both tumor and immune cells, complicate the accurate measurement of *PD-L1* expression in tumor tissues [8]. On the other hand, the possibility of measuring PD-L1 positive CTCs using the CellSearch system was a breakthrough in personalized therapy [9] since CTCs frequently express PD-L1, and this constitutes a clinically useful and non-invasive method to assess PD-L1 status in real-time [9,10].

## 2. CTC Interaction with Other Cells in the Blood Stream

CTC complex interaction with the immune cells of blood could provide a better understanding of the molecular pathways that are involved, leading to the improvement of therapeutic drugs and reduction of mortality and morbidity associated with cancer [11,12,13]. The activation of platelets represents a critical biological mechanism for metastatic progression since they shield CTCs and protect them from the attack of natural killer (NK) cells or macrophages and facilitate extravasation [14]. Neutrophils assist the metastasis of CTCs and promote tumor development by initiating an angiogenic switch and facilitating the colonization of CTCs [15]. It has also been reported that the abundance of tumor-associated neutrophils (cTAN) in advanced cancer patients contributes to CTC survival by suppressing peripheral leukocyte activation [16]. Moreover, neutrophils represent an important constituent in the formation of CTC clusters [17]. Neutrophil–lung cancer cell interactions are likely to be an important mechanism by which the progression of early malignancy is facilitated [18]. Dendritic cells (DC) cells also play a significant role in the formation of CTC clusters. Recently it was shown that DCs have a strong colocalization effect with CTCs [19]. Moreover, it has been shown that CTCs are associated with abnormalities in peripheral blood DCs in patients with inflammatory breast cancer (IBC). More specifically, IBC patients with ≥5 CTCs have low percentages and impaired function in both subtypes of DCs, indicating that immune cell profiling could add further prognostic value to CTCs in IBC patients [20].

Macrophages prime the premetastatic site and promote tumor cell extravasation, survival, and persistent growth. Macrophages are also immunosuppressive, preventing tumor cell attack by NK and T cells during tumor progression and after recovery from chemo- or immunotherapy [21]. In small cell lung cancer (SCLC), CTCs seem to recruit and “educate” a specific type of macrophages operative in the invasion, immune protection, extravasation, and possibly cachexia [22]. However, macrophages in the liver are major effector cells removing CTCs via antibody-dependent phagocytosis, an immune cell-mediated process preventing liver metastasis [23]. Cancer-associated macrophage-like cells (CAMLs), which are more frequent than CTCs, could provide complementary information for cancer detection and diagnosis [24]. Enumeration of CAMLs using the CellSearch system is related to worse progression-free survival (PFS) and overall survival (OS) compared to patients without CAMLs [25].

## 3. Clinical Significance of CTCs in Immunotherapy

### 3.1. Clinical Significance of CTCs and PD-L1^+^CTCs in Immunotherapy Using CellSearch Platform

CTC enumeration for MBC, metastatic prostate cancer (mPC), and metastatic colorectal cancer (MCC) using CellSearch technology enables the monitoring of cancer patients during therapy. Moreover, this technology captures and identifies tumor cells in the blood that are associated with poor clinical outcomes [26]. CΤC counts have also been associated in several studies with the prognosis of patients undergoing immunotherapy (Table 1). Alama et al. evaluated CTCs in 89 previously treated non-small cell lung cancer (NSCLC) patients receiving nivolumab. In this study, patients with baseline CTC numbers below their median values survived significantly longer [27]. Recent data have highlighted that the metabolic status could affect PD-L1 expression, such as PD-L1 degradation via mitochondria-associated oxidative phosphorylation inhibition [28]. In NSCLC treatment-naïve patients, CTC count variation (ΔCTC) was significantly associated with tumor metabolic response set by the European Organization for Research and Treatment of Cancer (EORTC) criteria. Moreover, elevated CTC count, along with metabolic parameters, were found to be prognostic factors for PFS and OS [29]. Tamminga et al. have shown that CTCs occur in one-third of advanced NSCLC patients, and their presence is of high prognostic and predictive value before and after immunotherapy [30]. In SCLC, that 5-year relative survival rate is extremely low; the use of a much higher cut-off equivalent to 150 CTCs/7.5 mL of whole blood also has clinical utility [31]. In a phase II multicenter adaptive immunotherapy trial of 457 longitudinal liquid biopsies from 104 patients with Metastatic Renal Cell Carcinoma (mRCC), the change over time of CTC enumeration is of prognostic importance [32].

An additional channel in the CellSearch system allows the examination of a fourth molecule of interest beyond the detection of cancer cells of epithelial origin. Establishment of the B7-H1/PD-L1 CTC analysis was performed for the first time by Mazel et al. showing that PD-L1 is frequently expressed on metastatic cells circulating in the blood of hormone receptor-positive, HER2-negative breast cancer patients [9]. This study was followed by several other studies that evaluated the PDL1 status of CTCs in various cancers using CellSearch (Table 1) [33,34,35,36,37,38,39,40].

Nicolazzo et al. showed that in NSCLC patients treated with the programmed cell death protein (PD-1) inhibitor nivolumab at baseline and at 3 months of treatment, the presence of CTCs and the expression of PD-L1 on their surface is associated with poor patients’ outcome. Moreover, 6 months after treatment, patients harboring PD-L1 negative CTCs obtained a clinical benefit, while patients with PD-L1^+^CTCs all experienced progressive disease, suggesting that the persistence of PD-L1^+^CTCs might mirror a mechanism of therapy escape [37].

**Table 1 biomedicines-11-01768-t001:** Clinical significance of CTCs and PD-L1^+^-CTCs in immunotherapy using CellSearch platform.

Type of Cancer	Number of Samples-Positivity	Additional Marker	Therapy	Response	Clinical Significance	Ref.
NSCLC	89 (91%); baseline	No	Nivolumab	n.a	Yes; OS (*p* = 0.05)	[27]
NSCLC	35 (45.7%); baseline 24 (41.7%); 8 weeks	No	Nivolumab or Pembrolizumab	Yes; tumor metabolic response (*p* = 0.004)	Yes; PFS (*p* < 0.001) OS (*p* = 0.024)	[29]
NSCLC	30 (36.7%); baseline	Yes; *PD-L1*^+^CTCs17 (11.8%)	Pembrolizumab	n.a	Yes; PFS (*p* = 0.034) OS (*p* = 0.023)	[36]
SCLC	21 (85.7%); baseline	No	Chemotherapy or chemotherapy/ immunotherapy	n.a	Yes; cut-off ≥ 150 CTCs/7.5 mL PFS (*p* = 0.02)	[31]
NSCLC	24 (83%); baseline 10 (67%); 3 months 10 (100%); 6 months	Yes; *PD-L1^+^*CTCs 20 (95%); baseline 10 (100%); 3 months 10 (50%); 6 months	Nivolumab	Yes; PD-L1-CTCs clinical benefit	n.a	[37]
NSCLC	53 (43.4%)	Yes; *PD-L1*^+^CTCs53 (9.4%)	ICIs	n.a	Yes; CTC countPFS (*p* = 0.006) OS (*p *< 0.001) PD-L1*^+^*CTCs OS (*p* = 0.002)	[33]
NSCLC	104 (32%); baseline 63 (27%); 4 weeks	No	ICIs	Yes; T0 (*p* = 0.02) T1 (*p* < 0.01)	Yes; baseline PFS (*p* = 0.05) OS (*p* < 0.01) 4 weeks (T1) PFS (*p* < 0.01) OS (*p <* 0.01)	[30]
NSCLC	39 (15.4%)	Yes; PD-L1^+^CTCs39 (33.3%)	ICIs	n.a	Yes; PFS (*p* = 0.040) OS (*p* < 0.001)	[35]
mPC	10 (50%); pre-ARSI 10 (50%); post-ARSI 10 (40%); mHSPC	Yes; ≥1 PD-L1^+^CTC10 (60%); pre-ARSI 10 (70%); post-ARSI 10 (40%); mHSPC	Abiraterone, acetate/prednisone or enzalutamide	n.a	n.a	[40]
MBC	124 (42%)	Yes;≥1 PD-L1^+^CTC 52 (40%)	Chemotherapy, endocrine therapy, targeted therapy	n.a	n.a	[34]
MBC	16 (100%); ≥1 CTC 16 (81.3%); ≥5 CTC	Yes; ≥1 PD-L1^+^CTC 16 (68.8%)	n.a	n.a	n.a	[9]
aUC	57 (47.4%); ≥1 CTC 57 (24.6%) ≥5 CTCs	Yes; ≥1 PD-L1^+^CTC 16 (62.5%)	Palliative systemic treatment	n.a	Yes; ≥5 CTC OS (*p* = 0.007)	[38]
MCC	51 (41%); ≥1 CTC 51 (33%); >1 CTC 51 (12%); ≥5 CTCs	Yes; ≥1 PD-L1^+^CTC 4 pts (<1% CTCs weak PD-L1)	n.a	n.a	Yes; ≥1 CTC OS (*p* = 0.030) >1 CTC OS (*p* < 0.020) ≥5 CTCs OS (*p* < 0.0001)	[39]

Sinoquet et al. have shown similar results concerning the worse outcome of PD-L1^+^CTCs, while PD-L1 expression in tumor tissue failed to prove any prognostic significance [33]. Apart from whole blood, different kinds of biological samples could be analyzed in a CellSearch analyzer, such as pleural fluid specimens. In NSCLC, the non-invasive measurement of PD-L1 expression in pleural EpCAM-positive cells (PECs), using the CellSearch^®^ technology, provides prognostic information and may improve the diagnostic accuracy of malignant pleural effusion (MPE) [41].

In mPC, immunotherapy against immune checkpoint inhibitors (ICIs) seems to be effective. For this purpose, identifying suitable biomarkers could facilitate the selection of the best candidates for this therapy [42]. Expression of PD-L1^+^ on CTCs in mPC patients during the administration of next-generation AR axis inhibitors is feasible and may enable monitoring of immunotherapy [40]. The expression of PD-L1 on CTCs in blood from patients with advanced urothelial cancer (UC) has also been analyzed. CTC detection and the presence of CTCs with moderate or strong PD-L1 expression are correlated with worse overall survival [38].

The assessment of PD-L1^+^CTC could also be applied in patients with Merkel cell carcinoma (MCC), which is a rare, aggressive skin cancer with increasing incidence and high mortality rates. Riethdorf et al. show that even though a high prevalence of CTC occurs at first blood collection that is associated with a worse prognosis, the overall frequency of PD-L1 production in CTCs is very low [39].

### 3.2. Prognostic and Predictive Value of PD-L1^+^CTCs in Various Types of Cancers

#### 3.2.1. NSCLC

In recent years, immunotherapy has become the first-line treatment for patients with NSCLC, with excellent responses in many patients [43]. However, many patients do not respond to this treatment, so the existence of biomarkers that can direct oncologists to appropriate treatment selection for each patient is essential. According to CheckMate 227, combined immunotherapy has demonstrated durable long-term efficacy benefits over chemotherapy in patients with advanced NSCLC and tumor PD-L1 expression greater than or equal to 1% or less than 1% across nonsquamous and squamous histologies [44]. However, apart from the analysis of PD-L1 in the tissue, its expression can also be studied in CTCs with proven clinical relevance (Table 2 and Table 3) [45]. Ilie et al. reported that PD-L1 expression in CTCs and circulating white blood cells obtained from 106 NSCLC patients correlated with the PD-L1 status in matched tumor-tissue samples [46]. Similar results were also reported by Abdo et al. in a comparative evaluation of PD-L1 in NSCLC patients showing good agreement rates on PD-L1 positivity (TPS ≥ 1%) and high PD-L1 expression (TPS ≥ 50%) [47].

Guibert et al. prospectively analyzed blood samples from 96 advanced-stage NSCLC patients obtained before nivolumab treatment and at the time of disease progression [59]. PD-L1 expression was more frequently observed in CTCs (83%) than in matched tissue samples (41%), and there was no correlation between CTC and tissue PD-L1 expression. Interestingly, a higher pre-treatment PD-L1 positive CTC number was observed in patients that did not respond to nivolumab [PFS < 6 months] [59]. In the same context, a previously mentioned study by Nicolazzo et al. included 24 patients with advanced NSCLC treated with nivolumab and assessed CTCs and CTC PD-L1 expression on blood samples obtained at baseline, 3 months, and 6 months post-treatment [37]. Although at baseline and 3 months post-treatment, detection of CTCs and PD-L1 positivity were associated with a dismal prognosis, at 6 months CTCs were found in all patients included. However, patients with PD-L1 negative CTCs continued to respond to immunotherapy, whereas patients with PD-L1 positive CTCs experienced disease progression, implicating that PD-L1 positivity on CTCs could be a predictive biomarker for early resistance to immunotherapy [37].

The examination of PD-L1 status through sequential biopsies could provide significant prognostic and predictive information due to status changes over time. A longitudinal evaluation of PD-L1 expression of CTCs isolated from NSCLC patients treated with nivolumab was reported by Ikeda et al. CTCs were enriched from 3 mL of peripheral blood using a microcavity array system at baseline and weeks 4, 8, 12, and 24 or until progressive disease. According to this study, PD-L1 expression on CTCs at week 8 has a superior predictive value compared to that at the baseline [55]. In this context, Moran et al. showed that at 18 months, patients showing an increase in PD-L1 expression had better clinical outcomes after ICI, with longer PFS (*p* = 0.0091) and OS (*p* = 0.0410) versus patients who did not demonstrate an increase in PD-L1 expression or the no ICI-treated population [48]. A longitudinal analysis was also performed in 47 advanced NSCLC patients receiving pembrolizumab. The results of this study revealed that changes in the PD-L1low subpopulation at an early phase of treatment are importantly related to disease control or resistance to pembrolizumab immunotherapy. Additionally, in patients with partial response, CTC counts were immediately increased at week 3, whereas the PD-L1low CTC rates were decreased [58].

PD-L1 expression presents heterogeneous expression in CTCs and tumor tissues from advanced NSCLC patients. Zhou et al. show that CTCs release a higher detection rate of PD-L1 expression than tumor tissues (53.0% vs. 42.1%). Moreover, NSCLC patients with PD-L1− on tissues but PD-L1^+^ on CTCs could still benefit from ICI therapy, while co-identification of PD-L1^+^CTCs or PD-L1^+^ tissues may help to identify patients who would benefit from immunotherapy [56].

Enough data support the fact that upon disease progression, NSCLC patients demonstrate an increase in PD-L1^+^CTCs, while no change or a decrease in PD-L1^+^CTCs is observed in responding patients [57]. Additionally, the increase of PD-L1^+^CTCs might indicate resistance toward PD-1/PD-L1 inhibitors. Similar results were shown by Sinoquet et al., where OS was significantly worse in NSCLC patients with PD-L1-CTCs and particularly in patients with PD-L1^+^CTCs compared with patients without CTCs. Moreover, the presence of PD-L1^+^CTC correlated with the absence of gene alterations in tumor tissue and with poor prognosis-related biological variables (anemia, hyponatremia, and increased lactate dehydrogenase) [33].

PD-L1 expression has also been studied in groups of patients receiving other types of treatment besides immunotherapy. Wang et al. studied gene expression of PD-L1 in CTCs isolated before, during, and after radiation or chemoradiation using a microfluidic chip. PD-L1 mRNA was highly expressed in patients who had disease progression within 9 months compared to those who had stable disease for 9 months or more, indicating that radiation therapy induces PD-L1 expression in CTCs [50].

The dynamic probability of PD-L1 as a surrogate marker has also been analyzed in multiple basket studies. Tan et al., in a study involving one hundred fifty-five patients with different advanced cancers, showed that the reduction in CTC counts and ratios of PD-L1-positive CTCs and PD-L1-high CTCs reflect a beneficial response to PD-1/PD-L1 inhibitors. In this study, patients with PD-L1-high CTCs had significantly longer PFS (4.9 vs. 2.2 months, *p* < 0.0001) and OS (16.1 vs. 9.0 months, *p* = 0.0235) than those without PD-L1-high CTCs [49]. Recently, a meta-analysis was reported, including results from 30 eligible studies (32 cohorts, 1419 cancer patients) about the prognostic significance of PD-L1 expression on CTCs in various cancers. The overall results from this meta-analysis showed that pre-treatment PD-L1^+^CTCs might predict better survival for patients receiving ICI treatment but worse survival for patients receiving other therapies. In addition, post-treatment PD-L1^+^CTCs were correlated with worse survival in cancers [60].

#### 3.2.2. HNSCC

In head and neck squamous cell carcinoma (HNSCC), the administration of immunotherapies has led to a response rate equivalent to 15–20%. However, ICIs have been approved for recurrent and metastatic (R/M) HNSCC patients as a first- and second-line therapy [61,62]. Recent data have revealed that CTC analysis is very promising in HNSCC [63,64,65]. However, studies on the clinical utility of PD-L1-positive CTCs are limited. In a prospective study including 23 HNC patients (Stages I–IV), PD-L1 status in CTCs was examined and correlated to patients’ survival. CTC enrichment was performed using the ClearCell FX system, which separates cells based on size (>14 µm) and deformability parameters. CTC immunophenotyping revealed that more than half of the patients (54.4%) appear to express PD-L1. Moreover, patients with CTC-positive counts had shorter PFS than patients with the absence of CTCs (hazard ratio [HR]: 4.946; 95% confidence interval [CI]: 1.571–15.57; *p* = 0.0063), and the PD-L1 positivity in the CTCs was found to be significant ([HR]: 5.159; 95% [CI]: 1.011–26.33; *p* = 0.0485) [51].

A highly sensitive, specific, and robust RT-qPCR assay for *PD-L1* mRNA expression in EpCAM(+) CTCs has been developed by Strati et al. for the detection of *PD-L1* overexpression in CTC. This prospective study enrolled 113 locally advanced HNSCC patients treated with curative intent at baseline, after two cycles of induction chemotherapy (week 6), and at the end of concurrent chemoradiotherapy (week 15). The findings of this study showed that patients with CTCs overexpressing *PD-L1* at the end of treatment had worse outcomes (PFS; *p* = 0.001, OS; *p* < 0.001), while its absence was strongly associated with complete response (95% CI = 2.76–92.72, *p* = 0.002) [10].

#### 3.2.3. Prostate Cancer

Immunotherapy represents a promising therapeutic option for the cure of prostate cancer patients [66]. In a phase II study, Boudadi et al. enrolled 16 patients with metastatic prostate cancer and AR-V7 expressing CTCs, that were prospectively treated with a combination of nivolumab and ipilimumab. Using targeted next-generation sequencing (NGS) in both pre-treatment tumor samples and CTCs, the authors found that high CTC phenotypic heterogeneity using the Shannon index was associated with improved response to combination immunotherapy. In addition, patients with defects in DNA repair genes (assessed by NGS in tumor biopsies or cell-free DNA in the case of no tissue availability) had higher CTC heterogeneity [67]. Zhang et al. performed CTC analysis for immune checkpoint ligands expression in men with mPC. Three cohorts of patients were enrolled, receiving different combinations of new-generation hormone therapy. High heterogeneity of immune checkpoint expression on CTCs was revealed across different disease states [40].

#### 3.2.4. Breast Cancer

Mazel and co-authors were the first to report the expression of PD-L1 on CTCs of patients with ER(+) HER2(−) breast cancer (BC) [9]. Interestingly, this study showed remarkable heterogeneity regarding PD-L1 expression in CTCs among the PD-L1 positive patients (11 out of 16, 68.8%). Schott et al. detected PD-L1 and PD-L2 expression on CTCs derived from blood samples of 128 patients with breast, prostate, lung, and colorectal cancer. In this study, patients with MBC had significantly more PD-L1 positive CTCs compared to patients with non-metastatic disease [68]. In addition, in one patient with MBC treated with combination immunotherapy (nivolumab/ipilimumab), the proportion of PD-L1 positive CTCs declined after the first and second dose of immunotherapy, whereas it increased following drug interruption, despite the persistently low level of CTCs.

Triple-negative breast cancer (TNBC) is an aggressive form of breast cancer that molecular targeted therapies are lacking. Immunotherapy has been included as standard care for stage II-III TNBC [69]. Vardas et al. studied a panel of ICIs, including PD-L1, in sixty-four BC patients with TNBC and thirty-one with luminal A or B of early and metastatic disease. Among BC subtypes, the phenotype of PD-L1^+^CD45^−^CK^+^ was higher in TNBC compared to luminal patients. Furthermore, among TNBC patients, there was an association of the phenotype PD-L1^+^CD45^−^CK^+^ with a shorter OS (7.6 vs. 53.8 months; log-rank *p* < 0.001, HR = 8.7) [54]

#### 3.2.5. Melanoma

The high immunogenicity of melanoma cancer makes immunotherapy one of the most effective treatment strategies [70]. Molecular characterization of circulating melanoma cells provides monitoring of the early response to immunotherapy [71]. Khattak MA et al. performed a longitudinal analysis of PD-L1 expression on CTCs in patients with metastatic melanoma receiving pembrolizumab prior to treatment and 6–12 weeks after initiation of therapy. PD-L1 positivity was prevalent in a high percentage of CTCs (64%) derived from melanoma patients. Moreover, patients with one or more PD-L1^+^CTCs had a higher response rate to pembrolizumab, as well as longer PFS compared with patients with PD-L1^-^CTCs (26.6 months vs. 5.5 months; *p* = 0.018) [52].

#### 3.2.6. Other Types of Cancers (Genitourinary Cancer, Bladder Cancer, Hepatocellular Cancer)

Chalfin et al. evaluated the T-cell counts and CTC morphologic features of metastatic genitourinary cancer patients receiving combination immunotherapy at baseline and on therapy at cycle 2 and cycle 3. Five distinct morphologic subtypes were identified by calculating the Shannon Index, and increasing CTC heterogeneity during therapy administration was associated with worse OS. Moreover, patients with CTCs > 4, specific CTC morphologic subtypes, PD-L1^+^, and low CD4 and CD8 T-cell counts had shorter survival [72].

Immunological response to bladder cancer is well conserved, and PD-L1 expression is differentiated between high-grade and low-grade cancers [73]. Morelli et al. show that 90% of non-muscle-invasive bladder cancer (NMIBC) patients have detectable CTCs, with a median CTC count of about four. A significant correlation between high PD-L1 and reduced recurrence-free survival (RFS) makes NMIBC patients’ ideal candidates for systemic approaches with ICIs [74].

In hepatocellular cancer (HCC), Su et al. investigated the predictive value of PD-L1 expression on CTCs in patients receiving PD-1 inhibitors combined with radiotherapy and antiangiogenic therapy. The count of PD-L1^+^CTCs was found to be an independent predictive biomarker of OS, and the objective response was more likely to be achieved in patients with a dynamic decrease in PD-L1^+^CTC counts at 1 month after treatment [53].

## 4. Immunotherapeutics on CTCs

CTCs acquire key properties required for metastatic spread and constitute an intermediate stage of metastasis [75]. They exist in the bloodstream as single cells or clusters of cells that are oligoclonal precursors of breast cancer metastasis [76]. The discovery of their molecular traits could facilitate the identification of targeted therapies [77]. Viable CTCs could be subjected to a dormant state through the immune-escape mechanism of CD47 upregulation [78,79]. Simultaneously blocking CD274 (programmed death ligand 1, PD-L1, or B7-H1) and CD47 checkpoints on CTCs by corresponding antibodies enhances the inhibition of tumor growth [80].

NK cells are of major importance in host immunity against cancer. Several different approaches to NK-based immunotherapy have been reported [81]. Allogeneic NK cells immunotherapy for recurrent breast cancer [82] and NSCLC [83] decrease CTC levels, which reflects the efficacy of treatment. A decrease in the number of CTCs is also an indication of oncolytic viral immunotherapy (Olvi-Vec). In an open-label phase 1b trial intraperitoneal, Olvi-Vec was given as monotherapy in two consecutive daily doses in 12 patients with platinum-resistant or refractory ovarian cancer. Immune activation was demonstrated from virus-enhanced tumor infiltration of CD^8+^ T-cells and activation of tumor-specific T-cells in peripheral blood, while at the same time, CTCs were diminished in 6/8 (75%) of baseline-positive patients [84]. A single-center prospective study demonstrated the short-term safety and efficacy of irreversible electroporation (IRE) combined with allogenic NK cell immunotherapy for unresectable primary liver cancer (PLCs). The combination therapy of IRE and NK cell immunotherapy significantly reduced CTCs and increased immune function and Karnofsky’s performance status. Moreover, PFS and OS were significantly improved in the IRE–NK group, demonstrating the synergistic effect of these two therapies [85]. Recent studies have also shown that exosomes derived from NK cells also exhibit antitumor properties. Kang YT et al. developed a streamlined microfluidic approach to on-chip biogenesis and harvest of natural killer cell-derived exosomes through comprehensive studies using NK cell lines and clinical samples from lung cancer patients. Circulating NK cell-derived exosomes have a cytotoxic effect against in-house patient-derived expanded CTC lines [86].

TAMs are the most frequent immune cells within the tumor microenvironment [87]. Sialic acid-modified EPI-loaded liposomes (EPI-SL) inhibit breast cancer metastasis by targeting TAMs and CTCs. A basic constituent of EPI-SL is the ligand of SA-CH, composed of sialic acid (SA) and cholesterol (CH). This is critical since SA-CH can directly bind to selectin, which is highly expressed on the surface of CTCs and effectively target and captures CTCs [88]. A HER2/neu vaccine-based immunotherapy for breast cancer has been reported in a pilot study by Stojadinovic A. et al. HER2/neu represents an attractive molecular target as an anticancer vaccine in breast cancer since it is overexpressed in up to 30% of breast cancers. E75^+^GM^−^CSF vaccination was applied in 16 patients with HER2/neu-expressing primary breast cancer, while thirteen of the 16 patients (81.3%) had at least one HER2/neu^+^CTC (mean: 2.1 ± 0.1 CTC/20 mL) in the peripheral blood. After vaccination, a reduction in CTC/20 mL (pre-vaccination 3.9 ± 1.5 vs. postvaccination 0.7 ± 0.4, *p* = 0.077) and HER2/neu^+^CTC/20 mL (pre-vaccination 2.8 ± 1.0 vs. postvaccination 0.5 ± 0.2, *p* = 0.048) was demonstrated [89].

In-vitro experiments have shown that immune activation of the monocyte-derived dendritic cells (Mo-DCs) using patients’ own CTCs is feasible. Kolostova K. et al. performed a co-culture of mature Mo-DCs (mMo-DCs) and autologous non-target blood cells (NTBCs). The activation effect of mature Mo-DCs on T-cell activation was monitored using multimarker gene expression profiling. Moreover, mMo-DCs might play a significant role in the PD-L1/PD1 regulatory axis since an elevated gene expression of PD-L1 was observed [90].

## 5. Future Perspectives and Conclusions

Liquid biopsy represents a novel, non-invasive approach for detecting and monitoring cancer through the analysis of its biological components, such as CTCs. The main challenge of the liquid biopsy era is the primary detection of minimal residual disease (MRD), where cancer cells, disseminated from the primary tumor, are non-detectable with conventional clinical or radiological tests, increasing the probability of new tumors formation of high metastatic potential. Sensitive and specific isolation and detection of CTCs is very important, especially in the case where surgical removal of the tumor is difficult. In this case, the information from the tumor cannot be available, and thus, oncologists do not have the proper guidance for the correct administration of targeted therapy to the patients.

Immunotherapy activates the body’s immune system to destroy cancer cells by enhancing the recognition ability of immune cells to the surface antigens of tumor cells, achieving their elimination. PD-L1 is a critical immune checkpoint protein that binds to PD-1 in T cells. ICIs are blocking the PD-1/PDL-1 interaction enabling immune system attack and sequentially destroying the cancer cells. That being said, it highlights the necessity of technologies that can accurately determine and assess the status of PD-L1 biomarkers and guide clinical oncologists as to whether cancer patients are suitable for immunotherapy. However, larger clinical studies are needed to be performed for the evaluation of the PDL1 status of CTCs and the integration of the PD-L1-CTC test into daily clinical practice.

## Figures and Tables

**Table 2 biomedicines-11-01768-t002:** Prognostic value of PD-L1^+^CTCs in various types of cancers.

Type of Cancer	CTC Isolation Technique	CTC Detection Method	Number of Samples (Positivity)	Therapy	Clinical Outcome	Ref.
NSCLC	CellSieve Microfiltration Assay	LifeTracDx PD-L1 test	30 (87%); low PD-L1 30 (13%); high PD-L1	ICIs	Yes; PFS-18 months (*p* = 0.0112) PFS-24 months (*p* = 0.0112)	[48]
Different advanced cancers	Pep@MNPs	IF	155 (81.9%)	ICIs	Yes; PFS (*p* < 0.0001) OS (*p* = 0.0235)	[49]
NSCLC	Graphene oxide (GO) Chip	IF and qPCR	38 (69.4%)	Radiation or chemoradiation	Yes; 5% cutoff (*p* = 0.017)	[50]
NSCLC	CellSearch	CellSearch	53 (9.4%)	ICIs	Yes; CTC count PFS (*p* = 0.006) OS (*p *< 0.001) PD-L1*^+^*CTCs OS (*p* = 0.002)	[33]
NSCLC	CellSearch	CellSearch	39 (33.3%)	ICIs	Yes; PFS (*p* = 0.040) OS (*p* < 0.001)	[35]
HNSCC	ClearCell FX system	IF	11 (54.4%)	Treatment naïve	Yes; PFS (*p* = 0.0485)	[51]
HNSCC	Ficoll–Hypaque density gradient	RT-qPCR	94 (25.5%); baseline 34 (23.5%); after IC 54 (22.2%); at the end of treatment	Chemotherapy	Yes; PFS (*p* = 0.001) OS (*p* < 0.001)	[10]
Various types of cancer	Pep@MNPs	IF	35 (74%)	PD-1 inhibitor IBI308	Yes; PFS (*p* = 0.002)	[49]
AM	Ficoll–Hypaque density gradient	Flow cytometric staining	25 (64%)	Pembrolizumab	Yes; PFS (*p* = 0.018) 12-month PFS (*p* = 0.012)	[52]
HCC	CytoSorter™ BioScanner system	CytoSorter™ CTC PD-L1 Kit	47 (48.9%); <2 PD-L1^+^CTC 47 (51.1%); ≥2 PD-L1^+^CTC	PD-1 inhibitor, IMRT, antiangiogenic therapy	Yes; OS (*p* = 0.001)	[53]
TNBC	Ficoll–Hypaque density gradient	IF	64 (41%)	Chemotherapy	Yes; OS (*p* < 0.001)	[54]
aUC	CellSearch	CellSearch	16 (62.5%)	Palliative systemic treatment	Yes; ≥5 CTC OS (*p* = 0.007)	[38]

**Table 3 biomedicines-11-01768-t003:** Predictive value of PD-L1^+^CTCs in various types of cancers.

Type of Cancer	CTC Isolation Technique	CTC Detection Method	Number of Samples (Positivity)	Therapy	Response to Therapy	Ref.
NSCLC	MCA system	MCA system	44 (82%); baseline 31 (58%); week 4 16 (56%); week 8 13 (62%); week 12 11 (55%); week 24	ICIs	Yes; *p* < 0.05	[55]
NSCLC	CellSieve Microfiltration Assay	LifeTracDx PD-L1 test	30 (87%); low PD-L1 30 (13%); high PD-L1	ICIs	Yes; PFS-24 months (*p* = 0.0091) OS-18 months (*p* = 0.0410)	[48]
NSCLC	Cyttel method	IF	117 (53.0%)	ICIs	No; prolonged mPFS-5.6 months (*p* = 0.519)	[56]
NSCLC	Parsortix system	IF	89 (56%); ≥1 PD-L1^+^CTC 89 (26%); ≥3 PD-L1^+^CTC	ICIs	Yes; Response (decrease or stable PD-L1^+^CTC) Disease progression (increase PD-L1^+^CTC) *(p* = 0.001)	[57]
Different advanced cancers	Pep@MNPs	IF	155 (81.9%)	ICIs	Yes; ORR (*p* = 0.018) DCR (*p* < 0.0001)	[49]
NSCLC	Ficoll–Hypaque density gradient	IF	47 (86%); baseline 43 (89%); after first cycle 23 (76%); after third cycle 19 (82%); PMR	Pembrolizumab	Yes; a decrease of PD-L1low CTC, partial response after the first cycle	[58]
AM	Ficoll–Hypaque density gradient	Flow cytometric staining	25 (64%)	Pembrolizumab	Yes; PD-L1^+^CTCs higher in responders (*p* = 0.005)	[52]
HCC	CytoSorter™ BioScanner system	CytoSorter™ CTC PD-L1 Kit	47 (48.9%); <2 PD-L1^+^CTC 47 (51.1%); ≥2 PD-L1^+^CTC	PD-1 inhibitor, IMRT, antiangiogenic therapy	Yes; <2 PD-L1^+^CTCs higher ORR (*p* = 0.007)	[53]

## Data Availability

Not applicable.

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
