# Peer review of "Clinical Significance of PD-L1 Status in Circulating Tumor Cells for Cancer Management during Immunotherapy"

_biomedicines, 2023, doi:10.3390/biomedicines11061768_

Round 1

Reviewer 1 Report

      In this research, the authors reviewed the current development of n Clinical significance of PD-L1 status in Circulating Tumor Cells for cancer management during immunotherapy. Generally, it’s meaningful and interesting review. In my opinion, the current version of this manuscript fits the scope of Biomedicines and could be accepted after major revision.

My specific comments are in detail listed below:

1.     Some minor mistakes exist in this paper, such as p < .0001. The authors should carefully check it.

2.     In this review, how immunotherapy affect the expression of PD-L1 in tumor cells including circulating tumor cells should be more detailed, especially chemo-immunotherapy. Some references should be added to this part including 10.1016/j.ijbiomac.2022.10.167.

3.     The abbreviations may be better listed in the front part of this paper instead of the tail part.

4.     Some minor mistakes exist in the references, such as ref. 29. The authors should carefully check and correct it.

5.     In this review, how the metabolic status affect expression of PD-L1 in tumor cells including circulating tumor cells should be more detailed, especially mitochondrial dysfunction. Some references should be added to this part including 10.1016/j.jconrel.2022.11.004.

6.     A more depth outlook or prospect that pointing out the future clinical therapy direction of using PD-L1 as a bio-marker in immunotherapy could be added.

Reviewer 2 Report

The authors presented an interesting review article topic about the clinical significance of PD-L1 status in circulating tumor cells for cancer management during immunotherapy. However, this review topic has been recently published by Tan et al. 2021, which negatively impacts the novelty of the review. I would recommend the authors use better-quality tables and figures that would enhance the overall quality of the review.

The paper needs to be written in a more scientifically appealing way to readers—taking care of grammatical and punctuation mistakes.

Round 2

Reviewer 1 Report

The current version of this manuscript could be accepted.